# Liver Metastasectomy for Metastatic Breast Cancer Patients: A Single Institution Retrospective Analysis

**DOI:** 10.3390/jpm11030187

**Published:** 2021-03-08

**Authors:** Armando Orlandi, Letizia Pontolillo, Caterina Mele, Mariangela Pasqualoni, Sergio Pannunzio, Maria Chiara Cannizzaro, Claudia Cutigni, Antonella Palazzo, Giovanna Garufi, Maria Vellone, Francesco Ardito, Gianluca Franceschini, Alejandro Martin Sanchez, Alessandra Cassano, Felice Giuliante, Emilio Bria, Giampaolo Tortora

**Affiliations:** 1Comprehensive Cancer Center, UOC di Oncologia Medica, Fondazione Policlinico Universitario A. Gemelli IRCCS, 00168 Rome, Italy; letiziapontolillo@gmail.com (L.P.); mariangelapasqualoni@gmail.com (M.P.); sergio.pannunzio90@gmail.com (S.P.); mariachiara.cannizzaro@outlook.com (M.C.C.); claudiacutigni1992@gmail.com (C.C.); antonella.palazzo@policlinicogemelli.it (A.P.); giovanna.garufi@unicatt.it (G.G.); alessandra.cassano@policlinicogemelli.it (A.C.); emilio.bria@policlinicogemelli.it (E.B.); giampaolo.tortora@policlinicogemelli.it (G.T.); 2Catholic University of Sacred Heart, 00168 Rome, Italy; maria.vellone@policlinicogemelli.it (M.V.); francesco.ardito@policlinicogemelli.it (F.A.); gianluca.franceschini@policlinicogemelli.it (G.F.); felice.giuliante@policlinicogemelli.it (F.G.); 3Comprehensive Cancer Center, Hepatobiliary Surgery Unit, Fondazione Policlinico Universitario A. Gemelli IRCCS, 00168 Rome, Italy; caterina.mele@policlinicogemelli.it; 4Multidisciplinary Breast Center, Dipartimento Scienze della Salute della donna e del Bambino e di Sanità Pubblica, Fondazione Policlinico Universitario A. Gemelli IRCCS, 00168 Rome, Italy; martin.sanchez@policlinicogemelli.it

**Keywords:** metastatic breast cancer, liver metastases, hepatic surgery, personalized medicine

## Abstract

The liver represents the first metastatic site in 5–12% of metastatic breast cancer (MBC) cases. In absence of reliable evidence, liver metastasectomy (LM) could represent a possible therapeutic option for selected MBC patients (patients) in clinical practice. A retrospective analysis including MBC patients who had undergone an LM after a multidisciplinary Tumor Board discussion at the Hepatobiliary Surgery Unit of Fondazione Policlinico Universitario “Agostino Gemelli” IRCCS in Rome, between January 1994 and December 2019 was conducted. The primary endpoint was overall survival (OS) after a MBC-LM; the secondary endpoint was the disease-free interval (DFI) after surgery. Forty-nine MBC patients underwent LM, but clinical data were only available for 22 patients. After a median follow-up of 71 months, median OS and DFI were 67 months (95% CI 45–103) and 15 months (95% CI 11–46), respectively. At univariate analysis, the presence of a negative resection margin (R0) was the only factor that statistically significantly influenced OS (78 months *versus* 16 months; HR 0.083, *p* < 0.0001) and DFI (16 months *versus* 5 months; HR 0.17, *p* = 0.0058). A LM for MBC might represent a therapeutic option for selected patients. The radical nature of the surgical procedure performed in a high-flow center and after a multidisciplinary discussion appears essential for this therapeutic option.

## 1. Introduction

Metastatic breast cancer (MBC) is the first oncological cause of death in women despite the advances in therapeutic strategies, with a 5-year survival of only ~25% [1,2]. The liver represents the first metastatic site in 5–12% of MBC [3] cases. Despite the transient response to chemo or endocrine therapy, most patients experience disease progression after 1–2 years [4]. While current evidence supports a liver metastasectomy (LM) for advanced colorectal cancer in improving survival [5,6] on the basis that hepatic parenchyma filters circulating tumor cells (CTC) from the primary neoplastic site to systemic circulation, LM is considered a possible therapeutic option for selected MBC patients in clinical practice, in the absence of prospective evidence. Several studies reported controversial results about the survival rate after hepatic loco-regional treatment in MBC with liver metastases with a 3-year and 5-year survival rate that ranged between 49–94% and 5–78% respectively [3,7,8,9,10,11,12,13,14,15,16,17,18,19,20,21,22,23,24,25,26,27,28]. A recent review of Bale et al. [29] showed that a primary tumor’s characteristics such as small tumor size, nodes negativity, low grade, and early-stage may be associated with a better outcome after liver surgery. In addition, they evidenced as an independent positive prognostic factor a long interval between the primary diagnosis and the detection of breast cancer liver metastasis (BCLM) more than 1 year, liver-limited disease (with the exception of isolated pulmonary and bone metastasis), response to preoperative systemic therapy before hepatic surgery, and the BCLM expression of estrogen receptor (ER) and progesterone receptor (PgR). The major limits of the studies in the literature are represented by the small number of patients enrolled and the presence of multiple confounding factors for the heterogeneity of the biology of the primary tumor, the presence of synchronous and metachronous metastases, the presence of extrahepatic disease, and the types of systemic treatments used. However, in all studies, patients with a low burden-disease benefited from R0 resections of BCLM with an improvement in survival rate [7,30]. Therefore, we report data about our experience of MBC patients who underwent liver metastasis surgery.

## 2. Materials and Methods

### 2.1. Study Design and Participants

A retrospective analysis including MBC patients who had undergone LM after a multidisciplinary Tumor Board discussion at the Hepatobiliary Surgery Unit of the Fondazione Policlinico Universitario “Agostino Gemelli” IRCCS in Rome, between January 1994 and December 2019, was conducted.

Eligible patients were aged 18 years or older, had a histological diagnosis of invasive BC and synchronous or metachronous LM. All immunophenotype BC were eligible in the study: luminal (ER and/or PgR positive), epidermal growth factor receptor 2 (HER2) positive and triple-negative (TNBC: ER, PgR, and HER2 negative). In all patients, disease assessment was determined by computerized tomography (CT) scan and magnetic resonance imaging (MRI) of the liver. The presence of extrahepatic disease was allowed provided that these sites were stable or in response to previous systemic treatments before hepatic surgery. The evaluation of expression of ER, PgR, and HER2 was done respecting the ASCO-CAP guidelines. Using the pathology report after hepatic surgery, the presence or the absence of disease at the resection margin (R0: no disease at the resected surgical margin, R1: the presence of disease at the resected surgical margin) was determined. For each patient, demographic data were collected including gender and age. Clinicopathological data on menopausal status (defined retrospectively after a woman has experienced 12 months of amenorrhea without any other pathological or physiological cause), metastatic sites, hepatic metastases presentation, number of systemic therapy pre-hepatic surgery, histotype (ductal *versus* lobular), immunophenotype, and resection margins were also collected. The study was approved by the Institutional Review Boards.

### 2.2. Study Endpoints

The primary endpoint was overall survival (OS) after LM, defined as the time from LM to death; the secondary endpoint was the disease-free interval (DFI) after LM, defined as the time from surgery to recurrence (in patients with liver-only disease) or progression of the disease (in patients with extrahepatic metastases which was stable or in response to previous treatment before LM). An exploratory analysis was performed to evaluate the survival impact of demographic and clinicopathological factors: age (<50 *versus* ≥50 years old), menopausal state (pre-menopausal *versus* menopausal), metastatic sites (only liver *versus* other), hepatic metastases presentation (synchronous *versus* metachronous), number of liver metastases (1 *versus* > 1), number of systemic therapy pre-hepatic surgery (none *versus* ≥ 1), histotype (ductal *versus* lobular), immunophenotype (luminal *versus* TNBC *versus* HER2+), and hepatic resection margins (R0 *versus* R1).

### 2.3. Statistical Analysis

Statistical analyses were performed using MedCalc software version 14 (MedCalc Software Ltd, Ostend, Belgium). Survival curves were calculated according to the Kaplan-Meier method and differences in survival were assessed with the log-rank test. Independent predictors of disease-specific survival and recurrence were identified by Cox proportional hazard analysis. Statistical significance was defined as a *p*-value < 0.05. As the study was explorative, an estimate of the sample size was not calculated.

## 3. Results

### 3.1. Demographic and Clinicopathological Characteristics of Patients

During the study period, a total of 49 patients, all female, underwent LM at our Hospital. Clinical data were available for 22. Patient age at the time of surgery ranged from 34 to 71 years with a median age of 48 years. Ten patients were premenopausal, 12 postmenopausal. Nineteen patients had isolated liver disease, 3 patients had multiorgan metastasis. Among patients with multi-organ metastasis, 2 had bone metastasis and 1 adrenal metastasis. Liver metastasis was metachronous for 17 patients and synchronous for 5 patients. Seven patients underwent surgery upfront, while 15 patients received one line of systemic treatment prior to surgery; the best response to systemic treatment was a partial response (PR) for 11 patients, 3 patients had a stable disease (SD), and only one had a progression disease (PD), with a disease control rate (DCR: PR + SD) of 93%. The histotype was ductal carcinoma for 21 patients, only 1 was lobular; 14 patients had a luminal tumor, 3 patients were HER2+, and 5 patients were TNBC. Nine patients underwent anatomical liver resection (resection of segments in 7 patients and resection of the left hepatic lobes in 2 patients were done) and 13 patients received metastasectomies (not anatomical liver resection). The resection margin was negative (R0) in 20 patients and positive in 2 patients. Among the 11 patients who had obtained a partial response, 4 patients had a pathological complete response (only fibrosis was found in the absence of neoplastic cells). Postoperative mortality (mortality within one month after hepatic surgery) was 0%. Complications occurred only in two patients: 1 patient presented perihepatic abscess and 1 patient with perihepatic abscess and a pulmonary embolism; both cases were resolved with medical therapy. All patients received at least one line of systemic therapy in the post-surgery setting: as maintenance of the previous treatment (hormonal therapy for luminal therapy, trastuzumab +/− hormonal therapy for HER2+ and the same chemotherapy in TNBC) and a new line of therapy after recurrence/progression of the disease.

Demographic and clinicopathological characteristics of patients are listed in Table 1.

### 3.2. Survival Outcomes

At the data cut-off analysis of May 2020, 11 patients were still alive and 7 patients were free of progression disease after hepatic surgery. Of the 15 patients who experienced recurrence, 8 have had disease progression with liver metastases, 3 with liver and bone metastases, 3 with lung metastases, and 1 with brain metastases. After a median follow-up of 71 months, median OS was 67 months (95% CI 45–103) (Figure 1) while median DFI was 15 months (95% CI 11–46) (Figure 2), respectively.

At univariate analysis, the presence of a negative resection margin was the only factor that statistically significantly influenced OS (78 *versus* 16 months; HR 0.083, *p* < 0.0001) (Figure 3) and the DFI (16 *versus* 5 months; HR 0.17, *p* = 0.0058) (Figure 4). None of the other factors were significantly associated with OS and the DFI; their association with the DFI and OS is shown in Table 1. A trend toward significance (the boundary of *p*-value < 0.2) was observed in the OS analysis for metastatic sites (only liver *versus* other sites, 103 *versus* 50 months, *p* = 0.14) while a prior systemic therapy showed a trend in favor also for the DFI (none *versus* ≥ 1, 14 *versus* 46 months, *p* = 0.1). The multivariate analysis confirmed the negative resection margin as the only factor which statistically significantly influenced OS (*p* = 0.0034) and DFS (*p* = 0.024).

Clinicopathological characteristics of patients with R0 resection are listed in Table 2.

Of the 20 patients with an R0 resection, 13 patients had a single lesion while 7 had two metastases. Radiological dimensions of the liver lesions are available for 13 of the 20 patients and ranged from 9 mm to 80 mm; histological dimensions were available for 15 patients and ranged from 4 to 35 mm. Fourteen patients had received one line of systemic treatment before the surgery: 4 patients had a complete response (CR), 6 patients had a partial response (PR), 3 patients had a stable disease (SD), and one patient experienced a progression of the disease (PD) before hepatic surgery; the DCR was 92.8%. Of about the 20 patients with an R0 resection, 19 had an immunophenotype of liver metastases consistent with primary tumor: 13 patients had a luminal immunophenotype (one of which was HER2 positive at diagnosis), 3 were HER2 positive, and 4 patients were TNBC.

Of the 2 patients with R1 resection, one had multiple (six) liver lesions and one a single metastasis, both were luminal consistent with primary tumor immunophenotype.

## 4. Discussion

Despite an improvement in the systemic treatment of MBC, the median survival of patients with metastatic disease is between 18 and 24 months [31]. Resection of breast cancer liver metastasis may represent a therapeutic option for selected patients. The radical nature of the surgical procedure performed in a high-flow center and after a multidisciplinary discussion appears essential for this therapeutic option [32] like in other neoplastic diseases [33]. 

In a recent systematic review of resection of MBC-LM, the median OS was 35.1 months and the median DFS was 21.5 months [34]. At the same time, in a case-matched analysis, the resection group had an impressive median OS of 82 months versus a median OS of 31 months in the systemic group, so the authors concluded that the combination of surgery with systemic treatment results in an improved OS [7].

The median OS and the DFI in our population were 67 months and 15 months respectively. Thus, our study seems to confirm a possible survival benefit in patients undergoing liver surgery of metastases especially in patients with an R0 resection. In fact, in our study, the presence of a negative resection margin was the only factor that statistically significantly influenced OS (78 *versus* 16 months; HR 0.083, *p* < 0.0001) and DFI (16 *versus* 5 months; HR 0.17, *p* = 0.0058). Of the 20 patients with an R0 resection, 13 patients had a single lesion while 7 had two metastases, this implies that a careful selection of patients with limited liver disease is important to obtain an adequate surgical result.

Fourteen patients received one line of treatment before surgery with a DCR of 93%; therefore, it also emerged in this evaluation that the selection of patients with a metastatic disease under control by systemic treatment can allow an important result. However, it is equally important to note that also the patients with PD during systemic therapy before liver surgery achieves an R0 resection, demonstrating how liver resection can also be proposed as a salvage treatment in highly selected cases. Moreover, surgical complications only occurred in two patients in the absence of post-surgery mortality, these data suggest that liver metastasectomy could be a safe procedure. At the same time, in our population, surgical radicality was achieved in almost all patients who were eligible for the study, and the R0 margin was found as the only prognostically relevant factor influencing both the DFI and OS. Taken together, these results confirm the importance of performing LM exclusively in high-flow centers, post a multidisciplinary discussion since, under this condition, the removal of liver metastases from breast cancer can significantly influence the survival of patients without significant side effects.

The main bias of the study is the smallness of the sample, also due to the limited availibility data of all the population of patients with MBC and undergoing liver surgery in our institution (49 versus 22). The sample size might have affected the results, limiting the statistical significance impact for some factors analyzed that appear to have a role in survival. In addition, the DFI and OS may have been influenced by the subsequent systemic therapies, but due to the heterogeneity of the population and the number and characteristics of treatment received post-surgery, it is not possible to evaluate their impact on survival outcomes. The trend benefit of the low tumor burden (only liver *versus* other sites) is in line with other results and with the suggestion of international guidelines to justify a multidisciplinary and more aggressive therapy in patients with limited metastatic disease in order to obtain a greater chance of healing [1]. The correlation with the menopausal state with better prognosis, on the other hand, could be related to a lower biological aggressiveness of the disease. Additionally, the trend benefit of the use of systemic pre-surgery treatment, as it has been shown in previous studies [3,8,18], seems to have a role in the increasing survival eradicating or debulking microscopic lesions. In contrast to other studies [25], in our population, we did not note an improved outcome for patients with luminal disease.

## 5. Conclusions

Despite the limitations imposed by a retrospective analysis on a small sample, our study confirms the possible positive role of R0 surgical excision of liver metastases from MBC if performed in a high-flow center after multidisciplinary evaluation. The prospective confirmation of this data appears to be increasingly necessary in order to consolidate the use of locoregional treatments in oligometastatic breast cancer disease, in particular, to identify the subgroup of patients who can benefit from surgical treatment. 

## Figures and Tables

**Figure 1 jpm-11-00187-f001:**
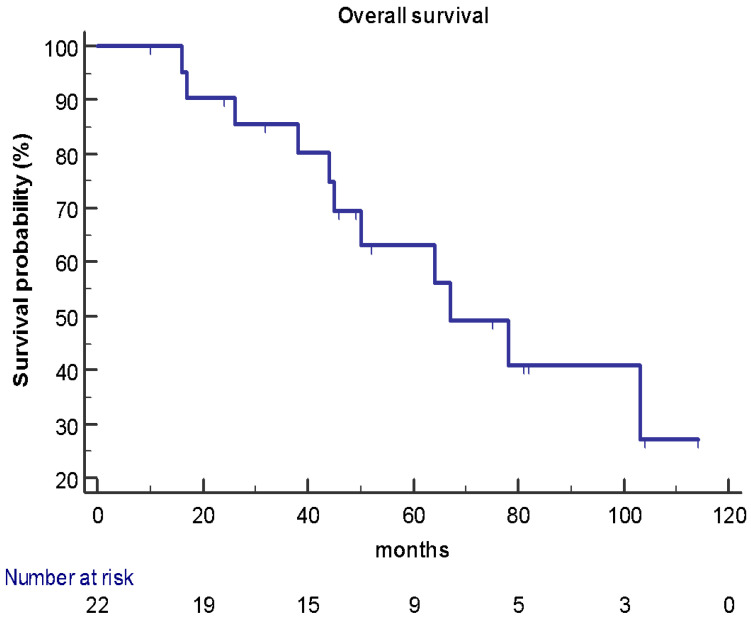
OS in the study population (*n* = 22): median OS was 67 months (95% CI 45–103).

**Figure 2 jpm-11-00187-f002:**
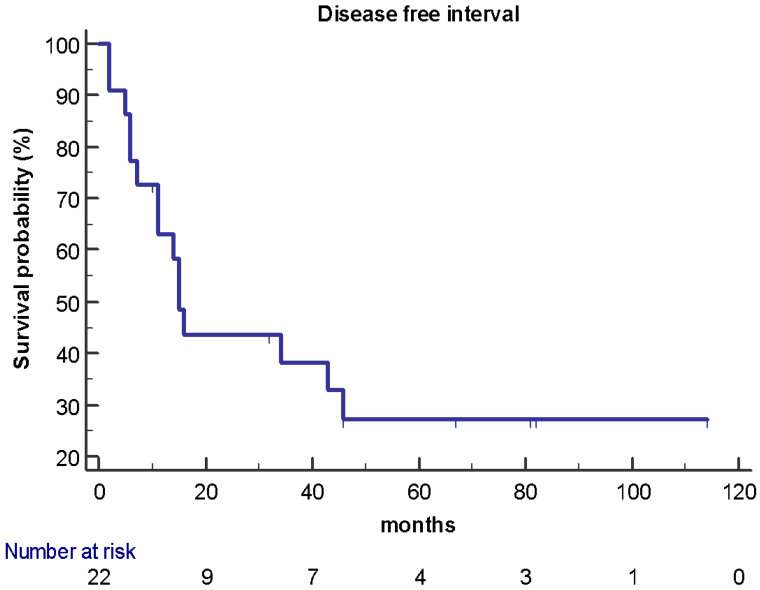
DFS in the study population (*n* = 22): median DFI was 15 months (95% CI 11–46).

**Figure 3 jpm-11-00187-f003:**
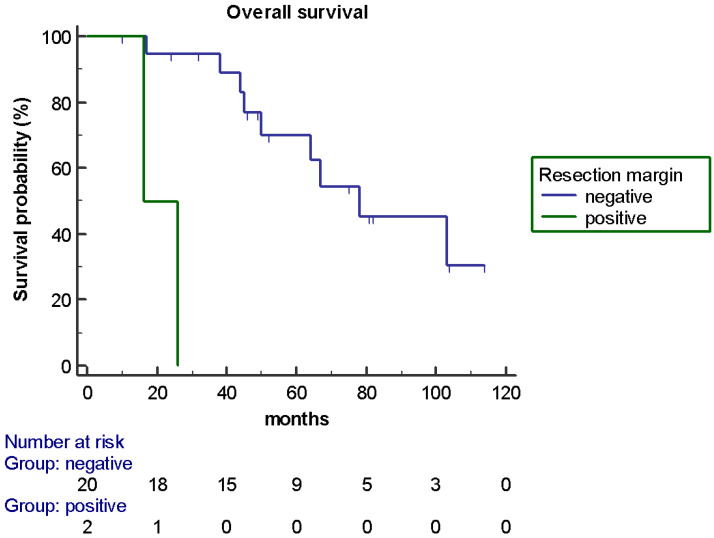
OS according to resection margin of liver metastases (R0 versus R1: 78 *versus* 16 months; HR 0.083, *p* < 0.0001).

**Figure 4 jpm-11-00187-f004:**
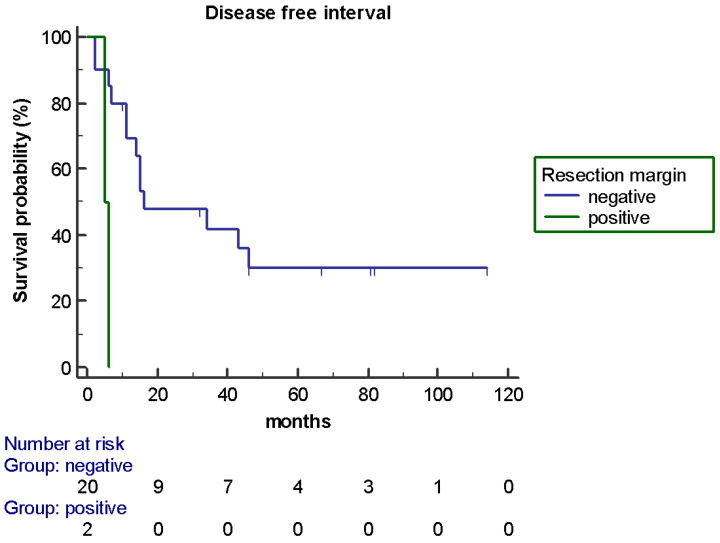
DFI according to resection margin of liver metastases (R0 versus R1: 16 *versus* 5 months; HR 0.17, *p* = 0.0058).

**Table 1 jpm-11-00187-t001:** Demographic and clinicopathological characteristics of patients (*n* = 22) and correlation with the disease-free interval (DFI) and overall survival (OS).

Characteristics	No. of Patients (*n* = 22)	DFS (Months)	Long Rank Test (*p* Value)	OS (Months)	Long Rank Test (*p* Value)
Age					
<50 >50	12 10	14 15	*p* = 0.7	50 103	*p* = 0.22
Gender					
Male Female	0 22	- 15	-	- 67	-
Menopausal Status					
Pre-menopausal Post-menopausal	10 12	14 15	*p* = 0.58	50 78	*p* = 0.56
Metastatic sites					
Only liver Other	19 3	16 11	*p* = 0.38	103 50	*p* = 0.14
Liver metastases					
Synchronous Metachronous	5 17	N.R. 15	*p* = 0.053	N.R. 64	*p* = 0.2
N. of liver metastases					
1 >1	14 8	15 11	*p* = 0.84	103 64	*p* = 0.52
Systemic therapy pre-liver surgery					
0 ≥1	7 15	14 46	*p* = 0.1	50 78	*p* = 0.89
Histotype					
Ductal Lobular	20 2	15 2	*p* = 0.88	78 44	*p* = 0.36
Immunophenotype					
Luminal HER2 + TNBC	14 5 3	13 17 7	*p* = 0.72	56 73 45	*p* = 0.28
Resection margin					
Negative (R0) Positive (R1)	20 2	16 5	***p = 0.005***	78 16	***p < 0.0001***

DFS: disease-free survival after liver resection; OS: overall survival after liver resection. N.: number. N.R.: not reached. Italics and bold: statistical significant *p*-value.

**Table 2 jpm-11-00187-t002:** Characteristic of patients with liver metastasectomy with negative resection margin (R0).

Characteristics	N. of Patients (*n* = 20)
Number of liver metastases	
1 >1	13 7
Systemic therapy pre-liver surgery	
0 ≥1	6 14
Best response of therapy before surgery PR SD PD	14 * 10 3 1
Immunophenotype	
Luminal HER2+ TNBC	13 3 4

N.: number. *: number of patients who received treatments before surgery. CR: complete response, PR: partial response, SD: stable disease.

## Data Availability

The datasets used and/or analysed during the current study are available from the corresponding author on reasonable request.

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
