# Peer review of "Liver Metastasectomy for Metastatic Breast Cancer Patients: A Single Institution Retrospective Analysis"

_jpm, 2021, doi:10.3390/jpm11030187_

Round 1

Reviewer 1 Report

Interesting results. in order to give more strength to these results we recommend to continue enrolling patients and to extend the number of patients. A single observation is to be done - the Ethical Committee approval for conducting this study should be included

Author Response

Interesting results. in order to give more strength to these results we recommend to continue enrolling patients and to extend the number of patients. A single observation is to be done - the Ethical Committee approval for conducting this study should be included.

Thanks for the comment and suggestion. We have tried to expand as much as possible the case series whose limited number represents the greatest bias of this retrospective study. However, it was complex, so we are evaluating to perform a combined analysis of the different studies in this setting, with a similar or lower number of patients enrolled, in order to obtain an analysis on a larger population. We confirm, as written in materials and methods, that the study has been submitted and approved by our local ethics committee (Line 96-97 “The study was approved by the Institutional Review Boards).

Reviewer 2 Report

The collection of such a serial data is not easy.  For this reason, the paper deserves such a credit for carrying out such a study.  Most of analyses are carried out in a very reasonable way.  I agree with the finding of the results may be important to the disease progression.

Author Response

Thanks for the encouraging comment.

Reviewer 3 Report

Liver metastasectomy for breast cancer patients is still being evaluated. The article is well written and will add to the growing evidence for this approach. The numbers are small as acknowledged by the authors. Careful patient selection but referral of all patients with liver metastases to a Liver MDM is essential. Early referral to Liver surgeons and continuing follow up in Liver MDM will enable more patients to be selected for surgery.

Author Response

Thanks for the comment, we share this careful reflection.

Reviewer 4 Report

Dear authors:

    I have two major suggestions about the prognostic factor and multivariate analysis. I also have several minor suggestions. These considerations are listed below.

Major suggestions:

  1. At line 100 and line 206 to 216, the author declares all 22 patients undergoing R0 or R1 resection for liver metastases. This implies that all hepatectomies were curative-intent, which also means no residual liver tumor after liver resection. The tumor number of liver metastases should also be included in prognostic factor analysis. It represents the severity of liver metastases. You may categorize as those in table 2.
  2. Between line 148 and 152, the author mentions about the trend toward significance. It is difficult to make a definition of significant trend, especially without sample size calculation in the retrospective study. Certain investigators consider a significant trend while p < 0.1, or sometimes while p < 0.2. Otherwise, it is always considered as not significant while p ≥ 0.2. In addition, while a significant trend is claimed, multivariate analysis must be done for adjustment.

Minor suggestions:

  1. The author cites reference 3 for the prevalence of breast cancer liver metastasis [He ZY, Wu SG, Peng F, Zhang Q, Luo Y, Chen M, Bao Y. Up-Regulation of RFC3 Promotes Triple Negative Breast Cancer Metastasis and is Associated With Poor Prognosis Via EMT. Transl Oncol. 2017 Feb;10(1):1-9. doi: 10.1016/j.tranon.2016.10.004.]. However, this study aimed to elucidate the influence of specific DNA expression (RFC3) on triple negative breast cancer, linking to lung metastasis. Other references must be cited to support the data of liver metastases.
  2. The reference 30 and 31 seem to be the same article. Does the author mean the article: Bale R, Putzer D, and Schullian P: Local Treatment of Breast Cancer Liver Metastasis. Cancers (Basel). 2019 Sep 11;11(9):1341 (doi: 10.3390/cancers11091341)?
  3. In the section Materials and Methods, what images for detecting liver metastases should be recorded.
  4. All these patients had metastatic disease, with or without extrahepatic sites. I suggest making a more accurate definition of disease-free/disease-recurrence in section Materials and Methods.
  5. According to the Kaplan-Meier curve of disease-free, 15 patients experienced disease recurrences. The recurrent patterns should be revealed in section Result.
  6. In table 1, the author divides the age groups by 50 years. World Health Organization uses 60 years for the definition of ageing [Integrated care for older people: guidelines on community-level interventions to manage declines in intrinsic capacity. Geneva: World Health Organization; 2017. Licence: CC BY-NC-SA 3.0 IGO.] Most investigators divide by older age, 65, 70, or 80 years. Is there any specific purpose of younger age?
  7. The definition of menopause is various in many studies [NCCN clinical practice guidelines in oncology, Breast Cancer, version 1.2021 January 15, 2021: page 74, BINV-O]. The author should describe the diagnosis of menopause more clearly in section Materials and Method.
  8. At line 115, the numbers of metachronous/synchronous of liver metastases seem error.
  9. At line 116 and 117, 4 patients had complete response after systemic treatment for liver metastases, which meant no detectable liver lesion by image study. How was the extent of liver resection decided while there was no detectable lesion?
  10. Between line 120 and 122, the liver resection extent was mentioned. I suggest adding more details of liver resection methods.
  11. Was there any systemic treatment after liver resection? It also may influence the survival. If there were, they should be revealed in section Result.
  12. At line 222 and 224, please unify the citation form in main text.

Author Response

Major suggestions:

  1. At line 100 and line 206 to 216, the author declares all 22 patients undergoing R0 or R1 resection for liver metastases. This implies that all hepatectomies were curative-intent, which also means no residual liver tumor after liver resection. The tumor number of liver metastases should also be included in prognostic factor analysis. It represents the severity of liver metastases. You may categorize as those in table 2.

Thank you for your suggestion. We added this assessment more clearly in the material and method (lines 114) and reported the prognostic analysis also in Table 1.

  1. Between line 148 and 152, the author mentions about the trend toward significance. It is difficult to make a definition of significant trend, especially without sample size calculation in the retrospective study. Certain investigators consider a significant trend while p < 0.1, or sometimes while p < 0.2. Otherwise, it is always considered as not significant while p ≥ 0.2. In addition, while a significant trend is claimed, multivariate analysis must be done for adjustment.

Thank you for your suggestion. We share the complexity of defining significance trends in a retrospective study without sample computation. We had arbitrarily considered a delta of OS> 20 months as a clinical significance trend. However, we took the suggestion to set the significant statistical trend boundary to p <0.2 and performed multivariate analysis as suggested (lines 174-198). Thanks again for this valuable consideration.

Minor suggestions:

  1. The author cites reference 3 for the prevalence of breast cancer liver metastasis [He ZY, Wu SG, Peng F, Zhang Q, Luo Y, Chen M, Bao Y. Up-Regulation of RFC3 Promotes Triple Negative Breast Cancer Metastasis and is Associated With Poor Prognosis Via EMT. Transl Oncol. 2017 Feb;10(1):1-9. doi: 10.1016/j.tranon.2016.10.004.]. However, this study aimed to elucidate the influence of specific DNA expression (RFC3) on triple negative breast cancer, linking to lung metastasis. Other references must be cited to support the data of liver metastases.

Thanks for this careful evaluation, we have properly corrected the reference.

  1. The reference 30 and 31 seem to be the same article. Does the author mean the article: Bale R, Putzer D, and Schullian P: Local Treatment of Breast Cancer Liver Metastasis. Cancers (Basel). 2019 Sep 11;11(9):1341 (doi: 10.3390/cancers11091341)?

Thanks for this careful evaluation, we have properly corrected the reference.

  1. In the section Materials and Methods, what images for detecting liver metastases should be recorded.

We have specified that all patients underwent a liver-specific contrast-enhanced magnetic resonance imaging (MRI) before hepatic surgery (lines 82-83).

  1. All these patients had metastatic disease, with or without extrahepatic sites. I suggest making a more accurate definition of disease-free/disease-recurrence in section Materials and Methods.

We have better specified the definition of disease-free interval (lines 107-110).

  1. According to the Kaplan-Meier curve of disease-free, 15 patients experienced disease recurrences. The recurrent patterns should be revealed in section Result.

We have added the description of the sites of disease recurrence as required (lines 166-168).

  1. In table 1, the author divides the age groups by 50 years. World Health Organization uses 60 years for the definition of ageing [Integrated care for older people: guidelines on community-level interventions to manage declines in intrinsic capacity. Geneva: World Health Organization; 2017. Licence: CC BY-NC-SA 3.0 IGO.] Most investigators divide by older age, 65, 70, or 80 years. Is there any specific purpose of younger age?

In consideration of the limited number of patients, we selected the age of 50, which corresponds approximately to the median age of the patients (48 years old), which allowed us to obtain two overlapping numerical samples to compare. At the same time 50 years old represent a common cut-off used in the interpretation of prognostic molecular tests in patients subjected to radical breast surgery (as OncotypeDx). In light of this, we have adopted this age limit.

  1. The definition of menopause is various in many studies [NCCN clinical practice guidelines in oncology, Breast Cancer, version 1.2021 January 15, 2021: page 74, BINV-O]. The author should describe the diagnosis of menopause more clearly in section Materials and Method.

Thanks for this suggestion. We have clarified (lines 101-102).

  1. At line 115, the numbers of metachronous/synchronous of liver metastases seem error.

Thanks for this evaluation. We have corrected the mistake.

  1. At line 116 and 117, 4 patients had complete response after systemic treatment for liver metastases, which meant no detectable liver lesion by image study. How was the extent of liver resection decided while there was no detectable lesion?

Thank you very much for this evaluation which allows us to report the results more correctly. Indeed, the 4 patients with “complete response” had a partial radiological response and a pathological complete response (no evidence of neoplastic cells in the area of metastasectomy: fibrosis only)(lines 139-141). Therefore, we have appropriately modified the results and Table 2

  1. Between line 120 and 122, the liver resection extent was mentioned. I suggest adding more details of liver resection methods.

Thanks for this evaluation. We added more details of liver resection (lines 136-139).

  1. Was there any systemic treatment after liver resection? It also may influence the survival. If there were, they should be revealed in section Result.

Thanks for this suggestion. We have added details as required (lines 143-147).

  1. At line 222 and 224, please unify the citation form in main text.

Thanks for this suggestion. We have corrected the citation.

Round 2

Reviewer 4 Report

Dear author:

    I think this article can provide well information of treatment in patients with breast cancer liver metastases.

Kind regards